# Innovative Collagen Based Biopolymers Tested as Fertilizers for Poor Soils Amendment

**DOI:** 10.3390/polym15092085

**Published:** 2023-04-27

**Authors:** Carolina Constantin, Daniela Simina Stefan, Ana-Maria Manea-Saghin, Irina Meghea

**Affiliations:** 1Department of Inorganic Chemistry, Physical Chemistry and Electrochemistry, Faculty of Chemical Engineering and Biotechnologies, University Politehnica of Bucharest, 1-7 Polizu Str., 011061 Bucharest, Romania; 2Department of Analytical Chemistry and Environmental Engineering, Faculty of Chemical Engineering and Biotechnologies, University Politehnica of Bucharest, 1-7 Polizu Str., 011061 Bucharest, Romania; 3Research Center for Environmental Protection and Eco-Friendly Technologies, Faculty of Chemical Engineering and Biotechnologies, University Politehnica of Bucharest, 1-7 Polizu Str., 011061 Bucharest, Romania; 4Department of Mathematical Methods and Models, Faculty of Applied Sciences, University Politehnica of Bucharest, 313 Splaiul Independentei Str., 060042 Bucharest, Romania

**Keywords:** fertilizers, collagen hydrogels, agrochemical tests, poor soils rehabilitation, application dose on soil

## Abstract

Improving soil quality is of growing interest and, among optimal solutions, the reuse and recycling of biopolymers of pelt waste from the tannery industry have been proposed, one of them being for collagen hydrolysate with micronutrients and polymers incorporated, to be used as fertilizers for poor soils rehabilitation. As functionalization agents, polyacrylamide, starch and dolomite were included into biopolymer matrixes in order to enhance their specific efficiency. These fertilizers were adequately characterized for their physical–chemical properties, including nutrient content, and tested on three poor soils, while a fourth sample of normal soil was chosen for comparative purposes. These soils were also characterized for their texture and physical–chemical properties in order to establish the fertility state of the soils as a function of nutrient content. In this respect, a series of agrochemical tests were developed at laboratory scale, simulating real agriculture environments in a vegetation room, where a significant plant growth in height was observed for all the agro-hydrogels with nutrients encapsulated, and multiplication of the nodosities number was observed in the case of the soybean culture. The most significant effect was obtained in the case of the fertilizer functionalized with starch. Finally, the application dose of the organic fertilizers for specific culture plants was estimated, such as field cultures (cereals, corn), field vegetables, vineyards or fruit-growing plantations. These agro-collagen fertilizers are particularly recommended for amendment of field cereals and vegetables. The novelty of this study mainly consists of the recovery and recycling of the pelt waste as efficient fertilizers after their adequate functionalization with synthetic or natural biopolymers.

## 1. Introduction

The continuous decline of soil fertility and crop productivity has resulted in an increasing interest to improve soil quality by adding various inorganic and organic amendments from different sources. A systematic study on using diverse biosolids in agriculture and their potential effects on soil and plant growth was reported by Sharma et al. [1]. Among the natural resources largely used for intensification of the bioproductivity of agricultural cultures, various natural zeolites, volcanic rock, gypsum and brown coal are mentioned [2,3,4], while biochar and straw have been applied in conjunction with chemical fertilizers [5]. Significant effects on growing soil fertility have been obtained by using organic composts with various types of waste, such as farmyard manure and crop residues [6], slaughterhouse waste [7], poultry litter [8], pig manure [9], spirulina from aquaculture waste [10] and micro-algal biomass [11]. Recently, an innovative poly-composite fertilizer was reported by recycling three organic wastes, sewage municipal waste, marine algae biomass and farmyard manure [12].

An alternative source to assure organic content is provided by the waste from the tannery industry, and significant research effort was devoted to produce protein composites with biochemical treatments using microorganism enzymes in order to obtain protein hydrolysates and protein binders with different uses. Organic biopolymers are a valuable source of raw materials for agriculture, as the protein waste matrix provides sufficient elements to improve the composition of poor and degraded soils, and many plants can benefit from the elements, such as carbon, nitrogen, calcium, magnesium, sodium and potassium [13,14].

Multicomponent, absorbent, hydrogel-type networks are next-generation materials with distinct three-dimensional structures and a high swelling capacity [15]. The applications of these materials have been diversifying in recent years, entering the fields of agriculture, food, pharmaceuticals, electrical devices and electronics [16], environmental protection and biomaterials.

Hydrolyzed collagen could be used in the food industry as a food supplement or in the cosmetic industry for skin care products [17,18]. By its functionalization with encapsulated nutrients, a product with applications as a biofertilizer in agriculture and horticulture has been obtained [19]. It is expected that such fertilizers will be cost-effective and available by recycling industrial units at prices at least five-times lower than similar marketable products, since there is no cost for the raw material. A quantitative estimation of the market cost of such fertilizers is in the range of 60–80 Euro/ton. Indeed, collagen can be recovered from pelt waste and used in various applications due to its high nitrogen content [20]. In this respect, the collagen obtained from wet blue leather waste after chromium extraction, for example, could be used as a nitrogen source and, after enriching with mineral potassium (K) and phosphorus (P), could be applied as NcollagenPK fertilizers for growing rice, soybean, sunflowers, maize, etc. [21,22,23,24], or as a potassium–phosphorus source from potato peel biochar [25].

Our previous research [26,27] developed new collagen composites produced by the incorporation of both mineral (phosphorus and potassium) nutrients and various synthetic and natural polymers into the collagen hydrogel matrix that was extracted with pelt waste processing in order to be proposed as fertilizers for the treatment of poor soils.

The novelty of the present work is focused on testing these innovative collagen composites for their fertilization qualities during application on three poor soil samples of Greek origin. For comparative purposes, a fourth Romanian normal soil sample is also tested. These soil samples are characterized with physical–chemical analyses, and the corresponding textural classes are assessed with specific granulometry measurements.

## 2. Materials and Methods

### 2.1. Materials

The synthesis of multi-polymeric, collagen-based agro-hydrogels was previously presented in our papers [26,27]. They were prepared using, as raw material, the limed hide waste (no haired) from the fleshing and trimming of bovine hides (lime fleshing) provided by SC PIELOREX tannery, Jilava, Ilfov county, Romania. Briefly, the gelatin hide obtained from collagen hydrolysate was further subjected to acid hydrolysis at 86–90 °C with 2.7–3.6% H_2_SO_4_ for 1.5–3.5 h in the presence of a K_2_HPO_4_ solution of 10–20% concentration, thus obtaining a reference fertilizer, denoted Ref-CH. In parallel, a polymeric solution of starch or polyacrylamide gel was prepared by continuous mixing, degasing and the addition of N,N’-methylene bis-acrylamide as the reticulant agent. Finally, the protein hydrolysate containing mineral nutrients encapsulated (phosphorus and potassium) was then functionalized with 5% of either a synthetic polymer—poly-acrylamide—resulting in the composite denoted POLY, or a natural polymer—starch—obtaining the composite STAR. In this way, we could compare the efficacy of a natural polymer, starch, with that of a synthetic polymer on the fertilization process. Another analysis was performed that compared the fertilization action of the two wastes, namely the organic biopolymer and starch, with a mineral additive, dolomite (prepared by adding 5% of dolomite suspension to Ref-CH), and thus obtaining the composite denoted DO. In this respect, while Ref-CH provided N-collagenPK nutrients only, the other additives incorporated into the collagen matrix provided supplemental contributions of nitrogen and organic carbon in the case of POLY, of organic carbon in the case of STAR, and of calcium and magnesium in the case of DO.

The plants selected for the tests on the poor soils were corn, sunflower and soybean for the following reasons.

Corn has good resistance to dry weather and many diseases and pests, good adaptability to various climate conditions and a good efficacy in the uptake of both organic and mineral fertilizers. The corn cob is a caryopsis containing starch, carbohydrates, protein substances, vitamins of the B group, E vitamin, iron, phosphorus, magnesium, zinc, potassium and oils.

Sunflower is one of the most widespread oleaginous plants; Romania occupies the fourth worldwide place after Ukraine, Russia and China. The sunflower seeds contain 50–54% fats, which are largely industrialized as oils, while the waste is used as animal fodder.

Soybeans contain over 30% protein substances and 17–25% oils. Its main role is to ameliorate the physical features of the soil as a result of installing symbiosis between the reticular system and nitrogen-fixing bacteria. In this way, many nodosities are developed to enhance the atmospheric nitrogen uptake and thus confer increased soil fertility.

### 2.2. Physical–Chemical Analyses of Soils and Harvested Crops

The determination of the particle size fractions of the soils was carried out with the pipette method for fractions < 0.002 mm, the wet sieving method for fractions 0.002 to 0.2 mm and the dried method for fractions > 0.2 mm [28].

The pH was determined with the potentiometric method in aqueous suspension of soil: water ratio 1:2.5 with a glass–calomel combined electrode; the potentiometer is calibrated with buffer solutions of known pH prior to the analysis of samples.

The organic matter (humus), OC%, was determined with the wet oxidation-modified Walkley–Black method.

The total nitrogen (TN%) was determined with the Kjeldahl method: digestion with H_2_SO_4_ at 350 °C, potassium sulfate and copper sulfate catalysts.

Exchangeable (accessible, mobile) phosphorus (EP%) was determined with extraction followed by spectrophotometry (CINTRA 404 UV–VIS spectrometer) with molybdenum blue after reduction with ascorbic acid [28].

Exchangeable (accessible, mobile) potassium (EK%) was determined with extraction followed by flame photometry (Flame Photometer S-935).

The alkaline soil carbonates of the air-dried soil fraction (particle size < 2 mm) were estimated with the calculation of measured carbon dioxide from carbonates decomposed with hydrochloric acid (1:3)—STAS 7184/16–80.

The total phosphorus was estimated with spectrophotometry (CINTRA 404 UV–VIS spectrometer), and the total potassium was estimated with flame photometry (Flame Photometer S-935) [29].

The four soil samples were thoroughly investigated according to ecopedologic indicators that were determined based on physical–chemical analyses: granulometry, pH, organic carbon (OC), total nitrogen (TN), exchangeable phosphorus (EP), exchangeable potassium (EK), carbonates (CO_3_^2−^) and heavy metals content.

The chemical analyses of the soils are detailed in Table 1 and Table 2.

According to ecopedologic indicators [30], these four soil samples are classified as follows:All the four soils fall in the weak alkaline reaction class with low organic matter;S1-L and S2-CL are in the class with low total nitrogen contents;S3-SiCL and S4-SCL are in the class with very low total nitrogen contents;S4-SCL is in the class with low exchangeable phosphorus;S1-L and S3-SiCL are in the class with very low exchangeable potassium;S2-CL is in the class with low exchangeable potassium;S4-SCL is in the class with medium exchangeable potassium;S1-L, S3-SiCL and S4-SCL belong to the class with low content of calcium carbonate with values below 1%.

Analysis of the data from Table 1 reveals, for the three samples S1-L, S2-CL and S3-SiCL, a very low nutrient content, phosphorous being not detectable, while for the sample S4-SCL, the data are specific for a fertile soil having a higher NPK nutrient content.

In Table 2, the metal content in the tested soils is quantified with special reference to essential metal nutrients and potential toxic metals. For this last category, one may observe that the metal contents are below the limit of environmental requirements except for nickel and manganese, which are within the alert zone for the sensitive soils in the case of the clay loam sample S2-CL.

The definitions in accordance with Romanian regulations [31] are the following:-The sensitive use of lands is represented by their use for residential and leisure areas, for agricultural purposes, as protected areas or sanitary areas with restricted regime, as well as the land areas provided for such uses in the future;-The less sensitive land use includes all existing industrial and commercial uses as well as the land areas provided for such future uses.

The alert thresholds are the concentrations of pollutants in air, water, soil or in emission/discharges, that have the role of alerting the competent authorities to a potential impact on the environment and that trigger the additional monitoring and/or reduction in pollutant concentrations in emission/evacuations.

Total heavy metals (Cd, Co, Cr, Cu, Ni, Mn, Pb, Zn, As and Hg) were quantified with atomic absorption spectrometry (GBC 932 AA atomic absorption spectrometer) using a calibration standard solution for each determined metallic ion.

The texture of the soils has been examined and the type of them is obtained using the triangular diagram of the basic soil textural classes according to USDA particle sizes from [32], page 182, Figure 8.13 as the location inside a ternary system built with the granulometric fractions in the corners: amounts of clay, silt and sand. According to these data, the following textural classes are established: soil S1 belongs to textural class L—loam; soil S2 belongs to textural class CL—clay loam; soil S3 belongs to textural class SiCL—silty clay loam; and soil S4 belongs to textural class SCL—sandy clay loam. In line with these findings, our tested soils are further coded as follows: S1-L, S2-CL, S3-SiCL and S4-SCL. More information can be found in the Appendix A.

The leaves harvested from the tested crops (sunflowers, corn and soybean) were first dried at ambient temperature, and then the dehydration was finished in the oven followed by grounding to a powder and, finally, sampling for chemical analyses.

### 2.3. Evaluation of Exchangeable NPK Nutrients of Agro-Hydrogels in Soils

Soil that was poor in nutritive elements sampled from the Western Greece region was placed in an aluminum vessel and mixed with the agro-hydrogel as liquid, dried or gel in various final proportions: 1%, 5%, 10%, 30%, 50%. Portions of these mixtures were sampled at different time intervals (e.g., 1, 3, 7, 14, 21 and 30 days) and analyzed for ammonium leaching. Changeable ammonium was evaluated by elution of the soil mixture with a solution 2.0 M KCl, and ammonium nitrogen was determined with a modified Kjeldal method.

### 2.4. Experiments in the Vegetation Room

The agrochemical experimentations were carried out in the vegetation room belonging to the National Research and Development Institute for Pedology—ICPA Bucharest. The seeds of green pea, sunflower, corn and soybean were sowed during the springtime (end of March) in plastic vegetation pots containing 1 kg of the tested soils sampled from Greece (two types of poor soils with very low nutrient contents characterized according to the data shown in Section 3.1). Both soils (S1-and S3-SiCL) were pretreated (amended) with 5% agro-hydrogels (Ref-CH, POLY and STAR) 4 months before the cultivation for a good homogenization of the agro-hydrogels added with soil and for adaptability of the existing microorganisms in the soils. The soil humidity in the vegetation pots was ensured at a permanent 70% level of the field capacity. According to the testing methodology, three replicates (three plants per single replication for the beginning and one month after rising and growth) were ensured for each experimental variant. The experiment was finished in September.

### 2.5. Estimation of Dose for Application of Organic Fertilizer

Agrochemical tests were conducted in the vegetation room on various plant species by monitoring the parameters related to the chemical composition of the soils, chemical composition of the plants and composition of microflora in the soil. Finally, the stage of plant growth was evaluated in order to assess which type of crops was most appropriate for the different soils.

The optimal dose of organic fertilizers in ton/ha of cultivated terrain was calculated based on the experiments in the vegetation room by applying a special formula and taking into account the humus content in the soil and the base saturation degree to evaluate the nitrogen index (IN), clay content and total nitrogen of the soil according to Equation (1) [29]:DOF, t/ha = (a + b/IN) × (c − d/Ag) × (e/Ng)(1)
where:DOF = Dose of Organic Fertilizer, t/ha;a, b, c, d empiric parameters with the following values:a = 15 for field crops and fruit-growing plantations; 20 for field vegetables and vineyards;b = 30 for field crops and field vegetables; 40 for vineyards; 50 for fruit-growing plantations;c = 1.35 for all cultures;d = 8 for all cultures;IN = nitrogen index of soil (% humus × V%)/100;Ag = clay content of soil (%);e = medium standard content of total nitrogen (%Nt) of the organic fertilizer = 0.45% from wet mass;Ng = total nitrogen content of the organic fertilizer, % from wet mass.

### 2.6. Field Experiments

Two of the fertilizers, S1-L and S3-SiCL, were incorporated into the soils, and the soil humidity was permanently maintained at 70% from the soil capacity. These tests were performed from the spring sowing until the autumn harvesting period.

### 2.7. Statistical Analyses

The agrochemical results represent the average (mean) of three replicates for soil and plant, respectively, and the standard deviation values {STDEV} have been calculated. The mean summarizes an entire dataset with a single number representing a typical value. It is one of several measures of central tendency. STDEV measures the amount of variability or dispersion, indicating how close the individual data values are from the mean value.

The factorial experimental design is also introduced, and the analysis of variance (ANOVA) is used as one of the primary tools for the statistical data analysis. There is a three-factor factorial experiment, the factors being the following: the first factor is for the considered four situations for the soils (no fertilizer soil, and the three soils treated with Ref-CH, POLY and STAR, respectively); the second factor is for two considered soils, i.e., loam soil S1-L and silty clay soil S3-SiCL; and the third factor is for the three cultivated plants taken into account in this discussion—corn, sunflower and soybean. Therefore, there is a total number of 4 × 2 × 3 × 3, the last factor representing the number of replicates.

## 3. Results and Discussion

### 3.1. Physical–Chemical Characteristics of the Fertilized Soils

The general evaluation of the fertility state of the soils was performed as a function of the soluble nutrient content available for plant growth [33], as shown in Table 3.

When these values are compared with the nutrient content of our soils, one may conclude that, for nitrogen content, the three soils, S1-L—S3-SiCL, are at the limit of very low fertility, and phosphorus is not detectable, while for potassium, only sample 2 corresponds to a good fertility state. That is why the new NPK agro-hydrogels proposed in this work could provide a promising source of nutrients to enhance the fertility state of these poor soils.

Three fertilizers were tested, including potassium phosphate incorporated as P–K nutrients source and considered as the reference sample together with its poly-composites obtained by functionalization with polyacrylamide, dolomite and starch. Their action on the soil samples was demonstrated by a series of agrochemical tests simulating real environmental conditions in the vegetation room for four culture plants. Finally, the dose of organic fertilizer to be applied for various agriculture crops in the field was estimated.

In Table 4, the results of the physical–chemical analyses for the fertilized soils are presented.

The average stem heights after one month from sowing of the three plant cultures are shown in Table 5.

From the analysis of these data, one may observe that, in all variants using agro-hydrogels with nutrients encapsulated, a significant plant growth in height was obtained, accompanied by a well-developed and profound reticular system, vigorous and healthy plantlets and multiplication of the nodosities number in the case of the soybean culture. The most important effect was produced by the agro-collagen hydrogel functionalized with starch (STAR).

These remarkable fertilizing effects are in good agreement with the monitoring results on the nutrients’ inclusion into the hydrogel matrix and their release towards the cultivated plants for the two tested poor soils (Table 4).

The mean values considered in Table 5 can be considered in good agreement with our experiment, since we discovered interactions between the variables as revealed in Table 6, which contains the ANOVA results.

The *F*-ratios for all three main effects and the interactions are formed by dividing the mean square for the effect of interest by the error mean square. Since we selected α = 0.05, the critical value for each of these *F*-ratios is *f*_0.05_(18,1) = 4.41. We could use the *p*-value approach. The *p*-values for all the test statistics are shown in the last column of Table 6. An inspection of these *p*-values is revealing. We observe that the most important factor is the treatment applied. In addition, the type of plant and the type of soil are significant. However, there is some indication of an effect due to the interactions of the treatments and soils, treatment and plants, and soil and plants, since the *p*-values are 0.056, 0.067 and 0.071, respectively, which are not much greater than α = 0.05.

### 3.2. Comparison of Functionalized Agro-Hydrogels Tested in Real Simulated Conditions

Agro-hydrogels containing starch and dolomite embedded into a collagen matrix with phosphorus and potassium nutrients incorporated were first tested on the normal soil S4-SCL for a green pea culture in order to ameliorate the field by soil enriching in biologically fixed nitrogen, which allows early clearing of the field to be further prepared for barley sowing. It is well established that a green pea culture needs nitrous fertilizers, mainly during the first development stage. Afterwards, this culture grows on account of the nitrogen uptake from the atmosphere through the bacteria formed in the root nodosities. The most appropriate soil reaction for a green pea culture is in the neutral pH domain, which requires amending the soils with lime, not directly, but by means of prior plant cultures.

In this respect, during the flourish phenological phase of the green pea culture, Diana variety, the number of nodosities on the roots were counted per plant on soil treated with 0.25–0.5 kg/m^2^. For the untreated soil, 17 nodosities per plant were counted; for the soil treated with hydrogel functionalized with starch (STAR), this number was 39; for dolomite (DO), this number was 37 (refer to Figure 1).

One may conclude that these collagen hydrogels functionalized with natural compounds have a high agrochemical efficacy, the best effect being obtained for the soil treated with the fertilizer containing starch.

Other agrochemical tests performed by using leather industry solid waste have been previously carried out for growth of the common bean [20] or rice [22], but only the effect of collagen as an organic nitrogen source has been investigated. Instead, our research is more complex, as these original smart fertilizers are able to provide all three main nutrients, NPK, while starch brings a supplementary contribution of organic carbon.

Another series of three agro-hydrogels with phosphorus and potassium nutrients incorporated, (Ref-CH)—as reference fertilizer, POLY—Ref-CH with polyacrylamide additive as the functionalization agent, and STAR—Ref-CH with starch incorporated, were extensively studied when applied on two Greek soils, S1-L—loam type and S3-SiCL—silty clay type soil, in order to increase their nutrient content.

As one can observe from the previous data of Table 3, the soils S1-L and S3-SiCL are characterized by a very low content of total nitrogen and extremely low content of exchangeable potassium, while the content in exchangeable phosphorus is not detectable. These experiments aimed at using the above-mentioned fertilizers on the two poor soils in order to stimulate the germination processes, accelerate the productive phase (plantlet growth), vigorously develop the plants, profoundly root the plants and strengthen the self-defense system of the plants.

The NPK content in the tested plants, sunflowers, corn and soybean, cultivated on the two Greek soils (S1-L and S3-SiCL) after one month from sowing, is quantified in Table 7.

A comparison of the monitoring results obtained on the treatment of these two poor soils showed that the nutrient leaching into the soil and plants was more efficient for the loam soil S1-L than for silty clay soil S3-SiCL, as expected [34], and the most significant effect was observed on soybean.

### 3.3. Estimation of Application Dose for Various Types of Culture Plants

This study demonstrated that the new agro-hydrogels belong to the category of organic GREEN fertilizers. In order to extend their applicative area, a complementary investigation was needed to select the most appropriate plant cultures with higher technical economic efficiency. Based on the experiments carried out in the vegetation room, the potential of application on various types of plant cultures, such as field cultures (cereals, corn), field vegetables, vineyards or fruit-growing plantations, was evaluated, and the optimal application dose by hectare was estimated using the Equation (1) defined in Section 2.5.

The results on organic fertilizer dose (DOF) in t/ha, estimated for the three agro-collagen hydrogels applied on the three poor soils, are listed in Table 8, Table 9 and Table 10.

Once again, we confirmed that the most efficient among the composite fertilizers tested was the collagen hydrogel functionalized with starch (STAR), and for the poor tested soils, the application dose varied between 25 t/ha on the loam soil cultivated with field cultures and over 92 t/ha for the silty clay soil cultivated with fruit-growing plantations. However, for economic reasons, these agro-collagen fertilizers are particularly recommended for amendment of field cereals and vegetables in the case of all poor soils tested.

## 4. Conclusions

This study focused on the systematic testing of the agrochemical features of three composite fertilizers based on collagen hydrolysate, incorporating both mineral nutrients (phosphorus and potassium) and natural or synthetic additives, denoted as follows: Ref-CH—collagen hydrogel matrix containing KH_2_PO_4_ encapsulated as the reference sample, POLY—Ref-CH functionalized with poly-acrylamide, and STAR—Ref-CH functionalized with starch.

In this way, we compared the efficacy of a natural polymer, starch, with that of a synthetic polymer on the fertilization process. Another analysis was performed by comparing the fertilization action of the two wastes, namely the organic biopolymer, starch, with a mineral additive, dolomite, denoted DO.

These composite fertilizers were tested in a vegetation room to follow the evolution of four plants from sowing to harvesting time: green pea, corn, sunflower and soybean. A comparison of the monitoring results obtained on the treatment of these soils showed that the nutrient leaching into the soil and plants was more efficient for the loam soil than for the silty clay soil, and the most significant effect was observed on soybean. In all variants using agro-hydrogels with nutrients encapsulated, a significant plant growth in height was obtained, and multiplication of the nodosities number was obtained in the cases of the green pea and soy cultures.

All the results were statistically analyzed based on the ANOVA test.

The most important effect was produced by the agro-collagen hydrogel functionalized with starch, which can be successfully applied in a dose of 25 t/ha for the amendment of all poor soils tested. Moreover, this biopolymer waste has the advantage of being cost-effective and biodegradable when applied on soil.

## Figures and Tables

**Figure 1 polymers-15-02085-f001:**
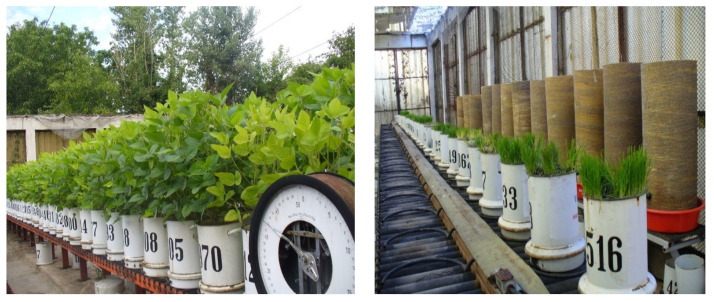
Monitoring plant cultures tested on poor soils in simulated field experiments.

**Table 1 polymers-15-02085-t001:** Chemical analysis of soils.

	pH	OC, %	TN, %	EP, mg/kg	EK, mg/kg	(CO_3_)^2−^, mg/kg
S1-L	8.30	2.41	0.12	Nd	50	37.5
S2-CL	8.27	2.81	0.11	Nd	128	-
S3-SiCL	8.50	2.28	0.09	Nd	42	54.3
S4-SCL	8.30	2.96	0.19	17	144	1.26

OC—organic carbon, TN—total nitrogen, EP—exchangeable phosphorous, EK—exchangeable potassium, Nd—below detection limit.

**Table 2 polymers-15-02085-t002:** Metal content in tested soils, mg/kg.

Soil Sample	Essential Nutrients		Potential Toxic Metals							
	Cu	Zn	Cr	Co	Ni	Mn	Pb	Cd	As	Hg
S1-L	47	82	53	15.4	107	940	16.4	0.233	5.54	0.032
S2-CL	63	108	63	24.8	139	1710	18.3	0.125	8.11	0.025
S3-SiCL	39	110	46	12.1	82	743	17.1	0.268	3.79	0.025
S4-SCL	32	64	48	10.8	22	978	8.67	0.087	0.18	0.010
Alertthreshold *	100/250	300/700	100/300	30/100	75/250	1500/2000	50/250	3/5	15/25	1/4

* Sensitive/less sensitive use of lands—types of land uses, which involve a certain quality of soils, characterized by a maximum accepted level of pollutants.

**Table 3 polymers-15-02085-t003:** General evaluation of the fertility state of the soil as a function of the soluble nutrient content available for plant growth.

Fertility State	Nitrogen (mg/kg)	Phosphorus, Aqueous Extract (mg/kg), P_2_O_5_	Potassium, Aqueous Extract (mg/kg), K_2_O
	NO_3_^−^ + NH_4_^+^		
Low to reduced	<10	<10	<50
Medium	10–30	11–20	50–80
High	>30	>20	>80
S1-L	12	Nd	50
S2-CL	11	Nd	128
S3-SiCL	9	Nd	42
S4-SCL	9	17	144

Nd—below detection limit.

**Table 4 polymers-15-02085-t004:** Physical–chemical analyses, mg/g of dried fertilized soils.

Parameters	S1-L	S2-CL	S3-SiCL	S4-SCL
pH	7.89	7.25	7.64	7.52
Temperature at sampling (°C)	2	4	10	14
Total phosphorous	0.012	0.011	0.015	0.014
PO_4_^3−^	0.040	0.042	0.044	0.043
Cl^−^	7.5	7.5	7.3	7.4
Cl_2_ free	1.0	0	0	0
NO_3_^−^	10	9.0	11	10
NO_2_^−^	0.02	0.02	0.03	0.05
S^2−^	0.021	0.017	0.028	0.029
SO_4_^2−^	0.021	0.016	0.028	0.030
CO_2_ free	5.17	2.53	1.54	2.2
NH_4_^+^	0.001	0.001	0.01	0.05
Total dissolved salts	30.8	33.2	36.9	38.0

**Table 5 polymers-15-02085-t005:** The average stem heights of the cultivated plants after one month from sowing (cm).

Experimental Variant	Loam Soil, S1-L			Silty Clay Soil, S3-SiCL		
	Corn	Sunflower	Soybean	Corn	Sunflower	Soybean
No fertilizer	20	18	15	18	16	13
Treated with Ref-CH	42	34	26	41	35	23
Treated with POLY	44	38	26	40	37	25
Treated with STAR	45	42	28	42	38	26

**Table 6 polymers-15-02085-t006:** Analysis of variance table for the three-factor, fixed-effects model.

ANOVA	(Balanced Design)		
Factor	Type	Levels	
Treatment	fixed	4	
Soil	fixed	2	
Plant	fixed	3	
Analysis of Variance for Stem Height			
Source	DF	F	P
Treatment	3	18.69	0.003
Soil	1	4.33	0.049
Plant	2	1.26	0.045
Treatment*Soil	3	3.10	0.056
Treatment*Plant	6	0.03	0.067
Soil*Plant	2	0.64	0.071
Treatment*Soil*Plant	6	2.08	0.188
Error	24		
Total	47		

* Represents interaction of the analysis factors.

**Table 7 polymers-15-02085-t007:** Nutrient content in plants cultivated on soils S1-L and S3-SiCL, mg/kg.

Agro-Collagen	Soil Not Treated			Ref-CH			POLY			STAR		
Nutrient	N	*p*	K	N	*p*	K	N	*p*	K	N	*p*	K
**Loam Soil S1-L**												
**Sunflower**	0.09	Nd	35	0.156	0.08	40	0.180	0.11	38	0.217	0.12	38
**Corn**	0.11	Nd	40	0.178	0.09	42	0.205	0.12	40	0.223	0.12	45
**Soybean**	0.12	Nd	42	0.216	0.11	44	0.245	0.14	41	0.246	0.13	48
**Silty Clay Soil S3-SiCL**												
**Sunflower**	0.05	Nd	28	0.142	0.07	35	0.168	0.09	35	0.203	0.10	37
**Corn**	0.08	Nd	35	0.152	0.08	37	0.189	0.10	37	0.214	0.11	42
**Soybean**	0.11	Nd	40	0.253	0.09	40	0.218	0.11	40	0.232	0.12	44

Nd—below detection limit.

**Table 8 polymers-15-02085-t008:** Estimated Dose of Organic Fertilizer, DOF, for Ref-CH agro-hydrogel in t/ha.

No Crt	Soil Sample		Field Cultures (Cereals, Corn)			Field Vegetables			Fruit-Growing Plantations			Vineyards Plantations		
			N	P_2_O_5_	K_2_O	N	P_2_O_5_	K_2_O	N	P_2_O_5_	K_2_O	N	P_2_O_5_	K_2_O
1	Soil 1 loam	DOF	36.79			37.80			87.41			71.55		
		NPK	0.82	0.50	0.79	0.84	0.52	0.81	1.95	1.20	1.88	1.6	0.98	1.54
2	Soil 2 Clay loam	DOF	46.46			47.64			110.84			90.56		
		NPK	1.03	0.64	1.0	1.06	0.65	1.03	2.47	1.52	2.39	2.02	1.24	1.95
3	Soil 3 Silty clay loam	DOF	55.08			56.24			132.47			107.83		
		NPK	1.23	0.75	1.19	1.25	0.77	1.21	2.9	1.81	2.85	2.4	1.48	2.32

**Table 9 polymers-15-02085-t009:** Estimated Dose of Organic Fertilizer, DOF, for POLY agro-hydrogel in t/ha.

No Crt	Soil Sample		Field Cultures (Cereals, Corn)			Field Vegetables			Fruit-Growing Plantations			Vineyards Plantations		
			N	P_2_O_5_	K_2_O	N	P_2_O_5_	K_2_O	N	P_2_O_5_	K_2_O	N	P_2_O_5_	K_2_O
1	Soil 1 loam	DOF	46.03			47.29			109.37			89.52		
		NPK	0.82	0.57	0.83	0.84	0.59	7.0	1.95	1.36	1.97	1.6	1.11	1.61
2	Soil 2 Clay loam	DOF	58.13			59.60			138.67			113.30		
		NPK	1.03	0.72	1.05	1.06	0.74	1.07	2.47	1.73	2.5	2.02	1.41	2.04
3	Soil 3 Silty clay loam	DOF	68.91			70.36			165.73			134.91		
		NPK	1.23	0.86	1.24	1.25	0.88	1.27	2.9	2.06	3.0	2.4	1.68	2.43

**Table 10 polymers-15-02085-t010:** Estimated Dose of Organic Fertilizer, DOF, for STAR agro-hydrogel in t/ha.

No Crt	Soil Sample		Field Cultures (Cereals, Corn)			Field Vegetables			Fruit-Growing Plantations			Vineyards Plantations		
			N	P_2_O_5_	K_2_O	N	P_2_O_5_	K_2_O	N	P_2_O_5_	K_2_O	N	P_2_O_5_	K_2_O
1	Soil 1 loam	DOF	25.60			26.31			60.84			49.80		
		NPK	0.82	0.37	0.68	0.84	0.38	0.70	1.95	0.89	1.62	1.6	0.73	1.33
2	Soil 2 Clay loam	DOF	32.33			33.15			77.13			63.02		
		NPK	1.03	0.47	0.86	1.06	0.48	0.88	2.47	1.13	1.67	2.02	0.92	1.67
3	Soil 3 Silty clay loam	DOF	38.33			39.14			92.19			75.04		
		NPK	1.23	0.56	1.02	1.25	0.57	1.04	2.95	1.35	2.45	2.4	1.1	1.99

## Data Availability

Not applicable.

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
