# Peer review of "Innovative Collagen Based Biopolymers Tested as Fertilizers for Poor Soils Amendment"

_polymers, 2023, doi:10.3390/polym15092085_

Round 1

Reviewer 1 Report

GENERAL COMMENTS

The author/s in this paper evaluate the Innovative collagen-based biopolymers tested as fertilizers for two poor soils amendment. This, together with the fact that the paper follows up on previous studies published in this journal, indicates that it is within the scope and is relevant to this Journal. The paper is well-written and presents an analysis of the factors affecting the uptake of biopolymers by crops, which is worthy of publication. However, there are some problems that the Author (s) need to address.

 DETAILED COMMENTS

Major Issues 1.

 The biggest omission in the paper is that there is no experimental design in the methodology which allow solving the uncertainties.

What design type?

What treatments?

How many experimental units? Randomization? And so son.

What were the study variables?

What Anova and test type were performed?

The authors need to organize the paper. For example, don´t mix the methods with the results

3. Abstract: It must be stated clearly at the outset what was the purpose of conducting this study.  Besides, include the experimental design, treatments, and variables. Results are also needed.

Minor Issues 1.

Keywords: Remove keywords duplicated in the title.

1. Page 3 Line 88 - 105: Remove all of them. It can be summarized and located in the Methods section.

2. Page 3 Line 107: Include the experimental conditions where the research was carried out

3. 2. Page 3 Line 109 - 127: Materials section must be summarized

 4. Page 3 Line 131 – 134: Paragraph must be redone. For example, corn crop has resistance to dry weather, diseases, and pests,…

5. Page 4 Line 145 – 167: Paragraph must be redone. Just cite the source of these methods.

6. Page 4 Line 212 – 218. Redo the paragraph and check the parallelism of verbs. It must be done throughout the manuscript.

7. Page 7 Line 260 - 211: Move the sentence “The average stem 260 heights after one month since sowing of the three plant cultures are shown in Table 4…”, to the Results section.   

7. Page 7 Line 260 - 211: Statistical analyses: include the ANOVA, Multiple range test, and the probability of the results.

8. Page 7 Line 278. The results and Discussion section must be REDO, based on experimental design.

9. Page 11 Line 380. REDO the conclusions based on stats and uncertainties.

Author Response

Dear reviewer,

Thank you for the careful reading and corrections made on my manuscript. All the changes suggested are reasonable, and I agree with them, as I consider that some of them really improve the article’s quality. The answers to these observations are given below in red. The response is attached.

Reviewer 2 Report

In this work, several collagen-based biopolymers have been prepared and applied as fertilizers in poor soil amendment. A series of agrochemical tests were used to evaluated their fertilizing effects for different crops. It was found that, agro-collagen hydrogel functionalized with starch exhibited remarkable fertilizing effects. Basically, some interesting results were reported. However, some modifications in this manuscript need to be done. I would recommend this manuscript accept after major revisions. The specific comments are as follows:

1. The language should be significantly improved.

2.The four soil types S1-L~S4-SCL first appear in Table 1, Line 91. However, the explanation of “L”, “CL”, “SiCL” and “SCL” was given at Section 3.1., Line 279. It may be beneficial for better understanding the differences of the four soil types to put this part of explanatory contents at the place that S1-L~S4-SCL first appear.

3.Table 4 and corresponding result analysis should be put in Section 3. Results and Discussion, rather than Section 2.6.

4. Some abbreviations, such as AMI, POLY, were not directly related to what it refers to. AMI cannot make readers thinking of agro-collagen hydrogel functionalized with starch. Appropriate abbreviations should be proposed.

5. The logical structure of the article needs to be optimized. For example, the contents of Section 3. Results and Discussion does not well correspond to the experiments in Section 2.

6. Lacking of discussion about Table 5.

7. Soil 1~ Soil 3 in Table 7-9 refer to which type of soil? Lacking of related explanations.

8. Abstract should highlight the novelty of this study.

Author Response

(The authors gave the same response as above.)

Round 2

Reviewer 1 Report

The paper was substantially improved. congrats.

Reviewer 2 Report

It can be accepted for publication.